# The Potential of Smart Factories and Innovative Industry 4.0 Technologies—A Case Study of Different-Sized Companies in the Furniture Industry in Central Europe

**Luboš Červený, Roman Sloup \*** and **Tereza Červená**

Faculty of Forestry and Wood Sciences, Czech University of Life Sciences Prague, Kamycka 129, 16500 Prague, Czech Republic
\* Correspondence: sloup@fld.czu.cz; Tel.: +420-608516302

**Abstract:** New innovative technologies of Industry 4.0 are the key to the future development of the furniture industry, which is outdated because of its atypical production and small-series production. For applying the novel trends of Industry 4.0 to the furniture sector, the methodical support of managers, the key users of these technologies, is essential. As there is a lack of knowledge regarding implementation of Industry 4.0, this study focuses on the evaluation of the current status of furniture companies in terms of production structure and Industry 4.0 benefits/threats with the aim of proposing methodological solutions for the implementation of this trend across different-sized enterprises. Data are collected using conduct-structured interviews with project managers who describe their own experience with Industry 4.0 implementation in central Europe. All interviews are analyzed using qualitative content analysis. According to the stakeholders, innovative production and non-production technologies are essential for their enterprises. Application of such technologies increases the efficiency of the whole operation by 30%–50% over the five years since the first innovations were introduced, especially in enterprises with atypical production and large enterprises. This study should serve as the tool for adapting the environmental changes and promoting the innovation approaches of the Industry 4.0 strategies on the central European level.

**Keywords:** forestry and wood sector; furniture technology; smart factory; project managers; furniture industry; Industry 4.0

## 1. Introduction

The European forestry and wood sector is currently undergoing significant economic changes that are due to the increasing pace of socio-economic and technical changes such as the climate crisis, the reduction in available natural resources, natural disasters, war conflicts and rising energy prices. In addition, consumers are demanding increasingly sophisticated products of high quality, standards and certifications, including support services, to meet their immediate needs.

This sector is unique in its interdependence within the raw material base, which should aim to develop the industries involved, processing capacities and the use of wood as an ecological and renewable raw material for future generations. Wood is of fundamental importance in this case, as it is used by the timber, furniture, paper and energy industries, followed by other downstream sectors such as construction. At the same time, this sector contributes to rural development from an economic, social and environmental point of view. This is also supported by the European Union's (EU's) strategic plan, "Agenda 2030" or "A Clean Planet for All" the new EU strategy for forests by 2030 [1,2], which highlights the need to identify the necessary changes to maximize the use of all energy, material and human resources involved in the value-creation process [3]. Businesses should be able to respond flexibly to these challenges with the latest innovations in their respective fields [4,5] through their value chains. By using both physical and virtual structures, close

collaboration and rapid adaptation can be achieved throughout the project and company life cycle, from production innovation to distribution innovation [6].

One of the tools for adapting to a changing environment is the introduction of information technology, cyber-physical systems and artificial intelligence systems into the production and services in all sectors of the economy [7–9]. The impact of these changes is so significant that we refer to them as the fourth stage of the Industrial Revolution [7]. This phenomenon was presented in Germany at the "Hannover Fair" in Hannover in 2011 as a proposal for a new concept of economic policy for Germany [10–13]. This direction of innovative technologies is characterized by the intelligent vertical and horizontal interconnection of people and machines [7,14–17] and objects and information and communication technology systems [17].

The main prerequisite for the implementation of this strategy is the innovation and modernization of production in the forestry and timber sector, where the main problems are considered to be the high cost of human labor, the outdated and worn-out production equipment and the lack of financial resources for further development. All of this can have a negative impact on the environment or on the sustainability of the whole sector [18].

The importance of technological development in this sector is underscored by the fact that, for example, in 2016 the government of the Czech Republic approved the Industry 4.0 Initiative prepared by the Ministry of Industry and Trade, which aimed to maintain and strengthen the competitiveness in the era of the so-called Fourth Industrial Revolution. In Europe, we can also encounter fully automated furniture plants where the human factor acts only as an additional member. An ecologically minded civilization requires changes in human thinking toward the harmonious coexistence between man and nature [19]. The rational use of environmentally friendly and sustainable renewable materials is a necessary step toward the development of an environmentally friendly industry [20–22].

While the available research shows the great potential of Industry 4.0 for business owners [23], its practical use is limited by a number of factors. One of them is the lack of knowledge and understanding of the potential of Industry 4.0 for different categories of businesses, mostly in atypical manufacturing [6,24].

It is clear that this vision will lead to increased complexity in manufacturing processes in the market at micro and macro levels. In particular, small and medium enterprises (SMEs) in the manufacturing industry are unsure of the financial requirements needed to acquire these new technologies and the overall impact on their business models [6].

Seminar experiences [25] on strategic orientation across enterprises have shown that the actors have serious problems in grasping Industry 4.0 across different concepts and operations. On the one hand, they are not able to link Industry 4.0 with their current corporate strategies; on the other hand, they are not experienced in identifying the state of corporate development and the vision of Industry 4.0. Therefore, they cannot identify specific areas for implementing the attributes of Industry 4.0. It is essential to formulate new models and tools in a way that they provide support with respect to the market needs of the sector and in line with the company's strategy.

For the successful implementation of Industry 4.0 in the furniture practice, it is essential to understand the importance of delivering the benefits of Industry 4.0, especially to project managers who are the key users of these measures. Therefore, the aim of this paper is to analyze what factors may influence the degree of the implementation of Industry 4.0 in the furniture industry depending on the size of the company and the type of production.

Other sub-objectives were:

1. To determine whether the implementation of Industry 4.0 depends on the size and type of production;
2. How the attitudes of project managers toward the introduction of new technologies depend on the size and type of production;
3. How the size of the company depends on the company's strategies;
4. How the risk of not implementing Industry 4.0 depends on the size and type of production.

From the answers obtained, a methodological framework for the implementation of Industry 4.0 will be proposed, which will be applicable to individual sizes of enterprises in the furniture industry and probably in other small-scale industries as well.

Sociodemographic characteristics or other aspects that may influence the subjective assessments of the Industry 4.0 implementation were also investigated. The assessment of attitudes and individual factors allow the identification of problems and obstacles, the solutions to which should then be incorporated in political decision-making and in the design of legislative, subsidy and information systems. These findings can contribute to shaping approaches and applying the objectives of national and European strategies and to the effective promotion of Industry 4.0 in this sector. Participatory methods could serve to increase the need for and consideration of Industry 4.0 in national strategic planning.

### 1.1. Literature Review

The rapid pace of technological improvements has created a need for rapid adaptation, and the most innovative companies are those that have been able to recognize early on how new digital tools are impacting their business models and what value they can derive from the information generated by their activities.

#### 1.1.1. The Current State of the Furniture Industry

The EU furniture industry, which consists mainly of SMEs, employs around 1 million European workers and produces around a quarter of the world's furniture, representing a market worth EUR 84 billion [26]. According to the classification of economic activities in the EU, furniture manufacturing falls under NACE (Nomenclature statistique des activités économiques dans la Communauté européenne) Division 31. The industry is fundamentally influenced by the customer, social trends, the cost of materials and labor, exports and competitive pressure or the situation on the commodities market. The furniture industry (NACE 31) is currently facing two main challenges. The customer, as a consumer, is looking for a quality product and professional services at the most reasonable prices and the increasing demands for environmental protection and the efficient use of natural resources in the context of the coronavirus pandemic. Smart manufacturing is an inevitable trend to maintain the viability of the NACE 31 sector [9]. Another important fact is that the wood products sector is the least automated industry on a European level—only 0.2% of the production processes in this sector are automated by robots involved in wood processing activities. The sector still performs most tasks manually and is characterized by a low understanding of the potential of automation. The average number of employees in this sector is decreasing at an annual rate of 1.8% [27].

#### 1.1.2. Industry 4.0 in the Procurement Sector

Digital transformation affects business models, production processes and corporate governance. In particular, improvements in information and communication technologies and analytical capabilities are spurring an influx of innovation at all levels of organizations. The opportunities offered by the use of digital technologies in the corporate world are changing the competitive position of companies and the way they interact with employees and customers, improving their market position [28].

The essence of Industry 4.0 is a comprehensive form of factory management that reduces all error factors. In such an environment, there is more efficient production and communication between people, machines and resources according to the principles of social networks [29,30].

The basis of Industry 4.0 is the technological development, advances in information and communication technologies and the internet, that connects the entire value chain [31]. Figure 1 presents the end of the Industry 4.0 concept. In short, the human–cyber–physical system (HCPS) technology enables real-time information acquisition, data analysis, strategic decision making and data processing. It leads to increased efficiency, logistics and a better demand response [32]. It is a completely new philosophy that brings about a societal

change that affects a large part of the industry, from technical solutions to work safety to the labor market or the social system. The cornerstones of Industry 4.0 (Figure 1) include the Internet of Things (IoT), Internet of Services (IoS), the HCPS [33,34], artificial intelligence (AI), big data and analytics, cybersecurity, augmented reality (AR), robotics, automation and visualization, cloud storage [10,35] and servitization [36].

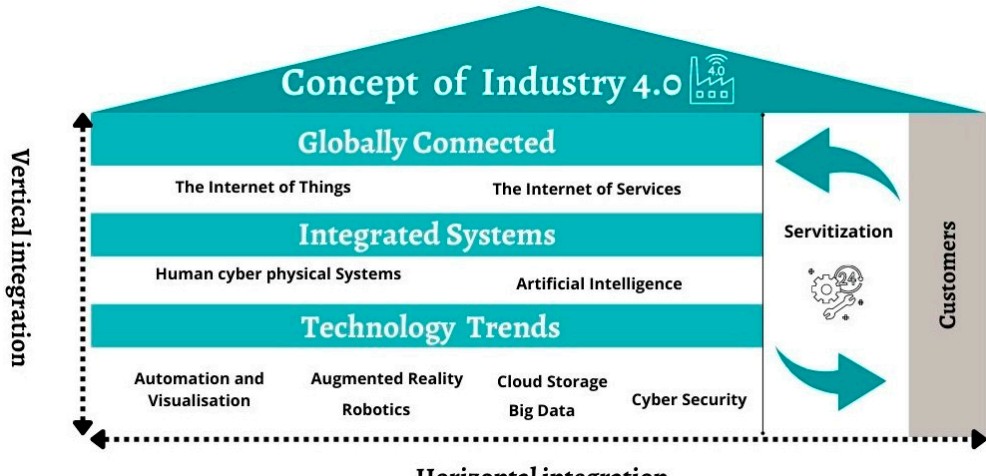

**Figure 1.** Scheme of Industry 4.0 integrating cornerstones.

The building blocks of Industry 4.0 represent a huge opportunity in terms of sustainability and increased productivity of industrial production and services, as well as the demand for skilled labor [37]. The coordination and integration of strategic and tactical operational decisions across the enterprise supply chain is essential to achieve all the Industry 4.0 objectives. Due to the complex and extensive challenges associated with the production process, strategic decision-making creates constraints in the tactical planning process.

The trade supply chain concept can help in planning decisions in complex industries [38,39]. The supply chain helps the production of raw materials to distribute products in the customers' regions, which helps the development of the micro-region. The forestry and timber supply chain consists of different process stages and different products such as biomass energy paper, semi-finished products for manufacturing and metal materials [40]. Another important level of decision-making in the supply chain is inventory management and planning, which, in an environment of uncertain supply and demand, accounts for 40% of the annual costs in the sector. The management system therefore requires coordinated decision-making on the inventory at all levels and at each facility in the supply chain. Such a supply chain is complex because it involves several units, each responsible for a large number of dependent activities [41].

## 2. Materials and Methods

The experiences and attitudes of project managers toward Industry 4.0 were investigated in several methodological steps, with an emphasis on a survey conducted through structured interviews and on the possibility of implementing these practices in practice. The interviews were conducted between 1 October 2021 and 1 June 2022.

The study was divided into the following steps:

1. Classification of the basic premises of the innovative technologies of Industry 4.0 and the creation of the structured interviews;
2. Pilot testing of the structured interviews;
3. Implementation of the structured interviews with the respondents;
4. Data processing, analysis and interpretation.

## 2.1. Step 1. Classification of the Basic Premises of the Innovative Technologies of Industry 4.0 and the Creation of the Structured Interviews

Based on the main assumptions of innovative Industry 4.0 technologies and the basic requirements for project management, the final version of the questions was divided into four thematic sections, namely:

- Attitudes toward the introduction of new technologies in the last two years, including any shortcomings in production;
- Potential for improvement in particular areas of the business;
- The company's strategy for the implementation of Industry 4.0 (obstacles, advantages, readiness for the implementation of innovative technologies and, if applicable, the financial resources used);
- Expected risks if Industry 4.0 is not implemented in the next five years.

The questions were compiled in collaboration with woodworking industry experts, IT and sociologists and served as a basis for the structured interviews with project managers dealing with innovative technologies in the woodworking industry. Specifically, this included companies producing furniture of various sizes and those developing Industry 4.0 software solutions.

Subsequently, enterprises were identified and classified according to the Commission Recommendation of 6 May 2003 concerning the definition of micro, small and medium-sized enterprises [42]. This breakdown was rather crude for the purposes of the research, so a more detailed breakdown of the enterprises (subcategories) was proposed and used in the research (Table 1), influencing the technology adoption in the enterprises according to the number of employees, turnover and type of production.

**Table 1.** Categories and subcategories affecting technology adoption.

| Category | Subcategories |
|---|---|
| Micro enterprises (up to 10 persons, turnover up to EUR 2 million) | - Up to 3 persons, turnover up to EUR 1 million<br>- Up to 10 persons, turnover up to EUR 2 million |
| Small businesses (up to 50 persons, turnover up to EUR 10 million) | - Up to 25 persons e, turnover up to EUR 5 million<br>- Up to 50 persons, turnover up to EUR 10 million |
| Medium-sized enterprises (up to 250 persons, turnover up to EUR 50 million) | - Up to 150 persons, turnover up to EUR 25 million (very untypical production)<br>- Up to 150 persons, turnover up to EUR 25 million (mass production)<br>- Up to 250 persons, turnover up to EUR 50 million |
| Large enterprises (more than 250 persons, turnover more than EUR 50 million) | - More than 250 persons, a turnover of more than EUR 50 million (built by gradual modernization)<br>- More than 250 persons, a turnover of more than EUR 50 million (built on a "green field") |

## 2.2. Step 2. Pilot Testing of The Structured Interviews

Pilot testing of the structured interviews with a test sample of stakeholders was conducted. The aim of the pilot collection was to test the logic and clarity of the questions. Based on the findings, several questions were subsequently refined.

## 2.3. Step 3. Implementation of the Structured Interviews with the Respondents

The actual testing (implementation of the structured interviews) was carried out on a total of 31 companies of different sizes in the wood processing industry in central Europe, specifically in Germany, the Czech Republic and Slovakia. The meeting was always pre-arranged in the company so that the intention could be sufficiently explained. The meeting took place either during a personal visit to the company or online via MS Teams. This solution was chosen primarily to allow contact with the respondents without the need

for a face-to-face meeting, mainly because of concerns about COVID-19 infections and measures resulting from Government Resolution 1375/2020 concerning restrictions on the free movement of persons [43].

Interviews with the project managers were recorded or transcribed. Subsequently, during a deeper analysis, the data obtained were transcribed and processed into a spreadsheet.

*2.4. Step 4. Data Processing, Analysis and Interpretation*

In terms of the number of questions answered, key areas and questions were selected for this paper, which were subsequently evaluated. Questions relating to the socio-demographics of the individual respondents and business data required to categories the businesses according to the suggested business sizes were also included. The responses obtained were then analyzed and a synthesis was made; the summary results were interpreted in the form of a table summarizing the main results in points. The tables were divided into four columns. The essence of the first column, "Example of a real company", is the formation of a certain spectrum of examples of companies, perceptible at the same time to a wide range of readers in a nonprofessional environment. The column "Proposed solutions" allows an understanding of the appropriate options for implementing the innovative technologies in each subcategory of enterprises. The column "Benefits of implementing Industry 4.0" provides a realistic view of the benefits of innovation for the enterprise. The column "Risks of non-implementation within 5 years" describes the possible threats that an enterprise may face if it is not implemented within a certain timeframe.

The advantage of this processing is the quick orientation in the results for the uniform categories of the company sizes and the possibility to easily find the solutions proposed to it. The benefit of this whole model is the direct applicability of the results to the furniture industry and the possible applicability of the results to other manufacturing sectors.

## 3. Results

A total of 94 respondents working in the furniture industry were contacted. Of these, 31 respondents were already involved in the implementation of Industry 4.0. For the purpose of this study, the focus was on data collected from this group of respondents ($n = 31$).

The socio-demographic characteristics of the respondents are shown in Table 2. The length of experience in the field was surveyed, with almost one-third of the respondents having 6–10 years of experience, and about one-fourth having 11–15 years of experience ($n = 10$).

The largest share of responses in terms of the region was dominated by respondents from companies operating in the Czech Republic (54.8%). Another question tracked the highest level of education attained. Overall, the highest number of respondents had a master's degree (41.9%), followed by a secondary education (25.8%), and 58% had a university degree. More than 77% of the respondents had a degree related to the furniture industry.

**Table 2.** Selected basic characteristics of the respondents.

| Data about the Respondent | | Is Your Company Involved in Industry 4.0? | | | | Total | |
|---|---|---|---|---|---|---|---|
| | | YES | | NO | | | |
| | | No. | % | No. | % | No. | % |
| | Total | 31 | 100.0% | 63 | 100.0% | 94 | 100.0% |
| Period of experience in the field | Less than 1 year | 0 | 0.0% | 9 | 14.3% | 9 | 9.6% |
| | 1–5 years | 7 | 22.6% | 12 | 19.0% | 19 | 20.2% |
| | 6–10 years | 10 | 32.3% | 19 | 30.2% | 29 | 30.9% |
| | 11–15 years | 9 | 29.0% | 14 | 22.2% | 23 | 24.5% |
| | More than 15 years | 5 | 16.1% | 9 | 14.3% | 14 | 14.9% |
| The company's market presence | Czech Republic | 17 | 54.8% | 27 | 42.9% | 44 | 46.8% |
| | Slovakia | 5 | 16.1% | 13 | 20.6% | 18 | 19.1% |
| | Germany | 6 | 19.4% | 14 | 22.2% | 20 | 21.3% |
| | Other | 3 | 9.7% | 9 | 14.3% | 12 | 12.8% |
| Highest education attained | Secondary education—leaving certificate | 5 | 16.1% | 20 | 31.7% | 25 | 26.6% |
| | Secondary school—high school diploma | 8 | 25.8% | 29 | 46.0% | 37 | 39.4% |
| | Bachelor's degree | 5 | 16.1% | 8 | 12.7% | 13 | 13.8% |
| | Master's degree | 13 | 41.9% | 6 | 9.5% | 19 | 20.2% |
| The link between education and the furniture industry | Yes | 24 | 77.4% | 49 | 77.8% | 73 | 77.7% |
| | No | 7 | 22.6% | 14 | 22.2% | 21 | 22.3% |

Table 3 shows that most of the respondents are in the small business category (38.7%), which corresponds approximately to their market share, as well as to the other size categories reported. The second most represented group are respondents from medium-sized enterprises (32.8%), followed by large enterprises (16.1%), where it is interesting to note that in enterprises with more than 250 employees, two-thirds of these enterprises have already implemented Industry 4.0. On the other hand, only 12.9% of the respondents in micro-enterprises have dealt with Industry 4.0 and only one-third of the respondents have experience with the implementation of Industry 4.0, which is partly due to the lack of interest in innovation and lack of financial resources that could be used for innovation.

**Table 3.** Number of respondents by size category of the enterprises.

| Data about the Respondent | Is Your Company Involved in Industry 4.0? | | | | Total | |
|---|---|---|---|---|---|---|
| | YES | | NO | | | |
| | No. | % | No. | % | No. | % |
| Total | 31 | 100.0% | 63 | 100.0% | 94 | 100.0% |
| **Micro-enterprises** | **4** | **12.9%** | **7** | **11.1%** | **11** | **11.7%** |
| Up to 3 persons, turnover up to EUR 1 million | 1 | 3.2% | 3 | 4.8% | 4 | 4.3% |
| Up to 10 persons, turnover up to EUR 2 million | 3 | 9.7% | 4 | 6.3% | 7 | 7.4% |
| **Small businesses** | **12** | **38.7%** | **28** | **44.4%** | **40** | **42.6%** |
| Up to 25 persons, turnover up to EUR 5 million | 5 | 16.1% | 12 | 19% | 17 | 18.1% |
| Up to 50 persons, turnover up to EUR 10 million | 7 | 22.6% | 16 | 25.4% | 23 | 24.5% |

**Table 3.** *Cont.*

| Data about the Respondent | Is Your Company Involved in Industry 4.0? | | | | Total | |
|---|---|---|---|---|---|---|
| | **YES** | | **NO** | | | |
| | No. | % | No. | % | No. | % |
| **Medium-sized enterprise** | **10** | **32.3%** | **24** | **38.1%** | **34** | **36.2%** |
| Up to 150 persons, turnover up to EUR 25 million (very atypical production) | 5 | 16.1% | 12 | 19.0% | 17 | 18.1% |
| Up to 150 persons, turnover up to EUR 25 million (mass production) | 3 | 9.7% | 8 | 12.7% | 11 | 11.7% |
| Up to 250 employees, turnover up to EUR 50 million | 2 | 6.5% | 4 | 6.3% | 6 | 6.4% |
| **Large Enterprises** | **5** | **16.1%** | **4** | **6.3%** | **9** | **9.6%** |
| More than 250 employees, turnover of more than EUR 50 million (built by gradual modernization) | 1 | 3.2% | 2 | 3.2% | 3 | 3.2% |
| More than 250 employees, turnover of more than EUR 50 million (greenfield) | 4 | 12.9% | 2 | 3.2% | 6 | 6.4% |

### 3.1. Implementation of Industry 4.0 in Micro Enterprises

This category includes a relatively large number of entities that mostly implement small projects that are not of interest to large entities, thus filling a gap in the market. Micro enterprises are not overly threatened by large companies, focusing mainly on local clients with specific customer requirements. The implementation of Industry 4.0 in micro enterprises is summarized in Table 4.

**Table 4.** Main results and recommendations (scored) for the micro enterprises and the subcategories (*n* = 4).

| Cat. | Subcat. | Example of a Real Company | Proposed Solutions | Benefits of Implementing Industry 4.0 | Risks of Non-Implementation within 5 Years |
|---|---|---|---|---|---|
| Micro-enterprise (up to 10 persons, turnover up to EUR 2 million) | up to 3 persons, turnover up to 1 million EUR | - Documents, information and tasks are transmitted directly;<br>- Small-scale contracts;<br>- Highly atypical production;<br>- Entities usually consist of one worker;<br>- Outdated machinery;<br>- Mostly hand tools. | - Cloud storage applications, cyber protection;<br>- Establishing a network of business and supply links;<br>- Work with data and 3D data. | - Speeding up the transfer of information;<br>- Efficient use of time;<br>- Increased work efficiency;<br>- Use of SME supply chain services;<br>- The enterprises will contribute indirectly to the development of the whole sector;<br>- Greater efficiency and growth. | - Micro-enterprises fill a gap in the market;<br>- They are able to address atypical customer requirements, which reduces the threat of competitive pressure from large players;<br>- The absence of innovation prevents progressive growth;<br>- The risks in the competition are minimal, the potential on offer is great. |
| | up to 10 persons, turnover up to 2 million EUR | - Family and individual companies;<br>- Atypical production;<br>- Limited spaces;<br>- High operating costs;<br>- Outdated technical and technological background;<br>- Creation of production documents of both groups mentioned above by "pencil and paper". | - Modernization of machinery;<br>- Innovation of non-productive parts of the company;<br>- For example, cloud storage, advanced data protection and mobility systems;<br>- If this is not under your own control, you need to look for entities providing professional services. | | |

The respondents report that, in the case of micro enterprises, data and information are transferred directly. Furthermore, these enterprises mainly carry out atypical small-scale production. Therefore, the objective of implementation is not complex automation and digitization. Rather than upgrading bulky machines, it is recommended to create a network

of suppliers who provide services in the form of cutting, gluing, milling, etc. Furthermore, the introduction of cloud storage would be useful.

The introduction of modern communication technologies will speed up the transfer of information not only between employees but also between suppliers and customers. The use of supply chain services by SMEs engaged in Industry 4.0 innovation will save time and processes on all sides and contribute to the development of the whole sector. These services will help the SMEs fill the gaps in their production. Micro enterprises are thus indirectly involved in the use of technologies that are oversized for them. At the same time, they can offer better processing and materials that they would not be able to process or details that they would not be able to produce. If individual operators understand this, the benefit will be to process more orders and secure more growth.

### 3.1.1. Up to 3 Persons, Turnover up to EUR 1 Million

Micro enterprises in this subcategory transfer information and tasks directly. They process small-scale orders with a focus on atypical production. The majority of these entities consist of a single manager and have basic, often outdated machinery. Large woodworking machines and equipment are often partially replaced by hand tools.

### Proposed Solutions

Cloud storage, along with cyber protection, are essential points for innovation in the smallest type of enterprise. Creating and leveraging a network of business–supplier relationships providing services in the form of cutting, gluing, milling, etc. is essential. It is thanks to cloud storage that the efficient sharing of production documents is made possible. A de facto option for a micro enterprise is to adopt the design software of the supplier company from which it receives services in the form of materials. By processing the 3D data by the contracting company itself, it is possible to create compatible documents that are shared with the interested company, on the basis of which it executes the services. This 3D data can be further used in communication with the customer, assembly line, etc.

### Benefits of Implementing Industry 4.0—For Both Subcategories of Large Enterprises

The respondents agreed that the implementation of Industry 4.0 will mainly speed up the transfer of information between employees and the supply chain. Cloud storage variants enable data sharing not only within the internal environment but also in the aforementioned cooperation with other entities. The effective use of modern technologies brings time savings, an increased work efficiency and an increased competitive advantage.

The use of supply chain services by SMEs engaged in Industry 4.0 innovations saves time and processes on all sides and contributes to the development of the whole sector. By using services, micro enterprises can indirectly participate in the development of the whole sector. They also reduce the price of raw materials and obtain a product that is processed with high quality on professional machines. This also contributes to processing more orders and ensuring greater growth.

### Risk of Non-Implementation within 5 Years—For Both Subcategories of Large Enterprises

Micro enterprises fill a gap in the market with their spectrum and flexibility. As they are able to address sophisticated and highly atypical customer requirements, they are not threatened by competitive pressure from large players. The absence of innovation will prevent effective development and a progressive competitive ascent.

The risks in the competition are minimal, but the potential it offers is great.

### 3.1.2. Up to 10 Persons, Turnover up to EUR 2 Million

These are mostly family and individual businesses, businesses with up to 10 employees engaged in atypical production, usually with limited space and high operating costs. The machines are often outdated and lack modern software to allow compatibility with other

equipment. The production documents of both these groups of micro enterprises are usually produced in an inefficient 'pencil and paper' way.

Proposed Solutions

The modernization of the machinery and the global innovation of non-production parts of the company are essential. Cloud storage, advanced data protection and mobility systems and engineering software are recommended. If the enterprise does not have expert in-house employees, it is necessary to seek entities providing IT services and other necessary services.

Micro enterprises should consider whether it is more efficient for them to produce furniture in-house or to have a range of cut, edged and milled semifinished products produced in collaboration with higher-end specialist operations that provide services as part of the trading and supply process. This potential is not sufficiently exploited or understood by businesses, and the production of a micro or small business can increase by multiples of the existing capabilities. As medium-sized enterprises become more technically and technologically equipped, these services become more affordable. By using services, the micro enterprise indirectly participates in development without fully addressing Industry 4.0.

### 3.2. Implementation of Industry 4.0 in Small Businesses

A small business with a long-term production spectrum is likely to feel the need to modernize and digitize its operations. If it understands the benefits on offer, the business will move toward greater efficiency. According to the respondents, the whole process of modernization is quite lengthy, bringing major changes in both the production and non-production parts that must be taken into account by the enterprises as summarized in Table 5, requiring modernization. It should be stressed that the process of technological change must be clearly elaborated upon with predetermined objectives and, above all, it must be carried out in steps that must always be perfectly executed.

### 3.2.1. Small Businesses, Sub-Category 25 Persons, Turnover up to 5 Million EUR

According to the characteristics of the respondents, these are companies that are undergoing a transformation from a small family-owned entity to a company with a wider range of contracts. Information concerning the production is distributed directly among the employees without the use of modern communication technologies. The production operation of cutting is carried out using conventional saws. The absence of modern banding machines is common in a company of this size. Outdated computer numerically controlled (CNC) centers are used where drilling and partial manual machining takes place. Processing of raw lumber is common in this category of company, which enters production as the main raw material in combination with the processing of agglomerated wood-based material.

Proposed Solutions

The development requires the creation and digitalization of the design and programming department, stock management, modernization of machines, and the division of labor of individual workers. The creation of work cells and departments that need to be gradually modernized and developed is required. Recording the time and material consumed directly at the workplace in a digital interface will streamline the entire business process. Introduction of an internal company network and distribution of data from designers or programmers to machine operators (shared storage with real-time data distribution). The design department becomes the heart of the furniture company, along with the networks that ensure the transfer of information.

**Table 5.** Main results and recommendations (scored) for small businesses and the subcategories (*n* = 12).

| Cat. | Subcat. | Example of a Real Company | Proposed Solutions | Benefits of Implementing Industry 4.0 | Risks of Non-Implementation within 5 Years |
|---|---|---|---|---|---|
| Small businesses (up to 50 persons, turnover up to 10 million EUR) | up to 25 persons, turnover up to 5 million EUR | - Transformation from a small family entity to a broad-based one; <br> - Companies often do not have departments such as production design and programming; <br> - The impossibility of integrating modern technologies; <br> - Verbal data distribution. | - Establishment and digitalization of the design and programming department, warehouse management, modernization of machines, division of labor of individual employees; <br> - Creation of work cells and departments; <br> - Digital time and material records; <br> - Implementation of an internal corporate network and data distribution. | - Real-time display of current data; <br> - Data availability; <br> - Increase in production capacity; <br> - Optimization of production costs; <br> - Automatic readers for registration and data collection, label scanning; <br> - The steps are provided by self-organizing structures. | - Unmodernized entities are unable to offer the required level of cooperation, are expensive and inefficient; <br> - Inefficient employees, lack of competence in Industry 4.0; <br> - Loss of customers, reduction in design quality. |
| | up to 50 persons, turnover up to 10 million EUR | - Corporate governance, a narrow circle of managers; <br> - Great efforts in management and communication; <br> - High diversity of technical and technological background and production processes; <br> - A combination of modern and obsolete machines; <br> - Combination of different input commodities; <br> - Companies feel the need to modernize, they don't have the know-how; <br> - The need to modernize spatial and technological facilities. | - Introduction of complex digitalization of the company; <br> - Design department, the digital twin of the product; <br> - Modernization of equipment; <br> - Cooperation with specialized IT companies <br> - Sophisticated management software, total personnel and project management; <br> - Creation of a position managing company development and employee training; <br> - Choice and specialization of processed commodity (solid sawn timber, wood-based panel material). | - Elimination of repetitive operations; <br> - The design department generates all data automatically, digitally; <br> - Applicable to all types of production; <br> - Specializing production in solid or plate material will streamline technical and technological processes; <br> - Freeing up space for underutilized technologies; <br> - Trade and supply links; <br> - -Increased internal transparency. | - In the absence of employees, there is a risk of paralysis of the enterprise; <br> - Without sophisticated software, there is a risk of information loss and business collapse; <br> - Failure to determine the direction of production will result in high operating and development costs; <br> - The spatial layout will limit the growth of the company. |

Benefits of Implementing Industry 4.0

Among the benefits of implementing Industry 4.0, the respondents cited the data collection and distribution to cloud storage or internal servers. Viewing real-time, up-to-date data outside the company allows for better governance and operational management. Data from production and non-production departments are used for future company growth and process optimization. Modern machines allow one to handle increased production capacities.

These devices include automatic readers for registration and data collection, as well as for the scanning of labels, contributing to an automated process in the company. These and other processes reduce the dependency on key persons and reduce the risk of downtime for these important persons in the running of the business. Thanks to the innovations, the company can offer its customers and business partners an interesting collaboration in terms of price and quality.

Data from the production and non-production departments that are recorded by machines and workers are collected in cloud storage. If the company cannot handle the data now, the data can serve in the future development of the company.

Risk of Non-Implementation within 5 Years

Businesses that do not invest in the development and retraining of workers are unable to offer the required level of cooperation and are expensive and inefficient. Untrained employees in Industry 4.0 often cannot work with basic data or design software. This discourages and terminates the possibility of collaboration with mature companies and reduces success in competitive development.

3.2.2. Up to 50 Persons, Turnover up to EUR 10 Million

A business of this size requires a great deal of effort in management and communication. At this level, non-modernized businesses are led by one or two key people, and their sudden absence often affects the entire existence of the company. Enterprises have a wide variety of technological equipment and production processes depending on the processing of input commodities (e.g., solid sawn timber and agglomerated board materials). We encountered plants that work with outdated machines or software, as well as companies that install state-of-the-art self-cutting centers, or combinations of both. Companies feel the need to modernize, but they do not have sufficient know-how and do not understand the potential of Industry 4.0. The requirement is directed toward modernization of the equipment, space and technological facilities.

Proposed Solutions

Comprehensive digitization of the preproduction part, i.e., all the data entering production from the design department, will streamline the entire production flow. Companies are advised to create a cyber twin of the product. A digital copy of the product will enable the data transfer and modification and simulation of the individual production phases. The company should consider a complete reconstruction of the software and hardware infrastructure of the enterprise, aiming at the complete interconnection of all sections of the company.

Establishing cooperation with specialized IT companies ensures the operation of the entire company. These specialist companies have tailored software solutions that connect and automate individual departments, reducing repetitive human resource activities. The software in place enables data to be flipped from accounting and design programs and to be seen in a global view by all people and project management.

Respondents suggest the creation of a job position ensuring the development of the company and the training of employees. They further state that where the job position has been created already, there has been an effective shift in the overall development of the company.

Benefits of Implementing Industry 4.0

The company's innovation in digitalization will eliminate repetitive operations that burden workers in all positions of the company. The design department generates all the digital data needed to implement a complex production flow by modeling the so-called "live model", automatically generating production documentation, drawings, material orders, CNC machining programs, parts carrying information about the material, edges, production dimensions, production process and more. The technology can be applied to batch plants, as well as to entities engaged in serial and highly atypical production, which is inherent in a company of this size.

According to the respondents, a change in access to the raw material input into the production flow is essential. Specialization of the production in the processing of raw sawn timber or board material is necessary to make all the processes more efficient. An example is the elimination of single-purpose machines for processing raw sawn timber.

The companies surveyed use partnerships with other companies to replace the in-house technologies for processing solid sawn timber that have been discontinued. These can, for example, supply solid wood semifinished products, thus creating business supply links.

The creation of a development department and a position responsible for development will ensure the effective implementation, training of staff and communication with all entities involved in the development.

Risk of Non-Implementation within 5 Years

Failure to implement these technologies leads to the exploitation of key employees whose potential is not effectively used. In their absence, the company is at risk of paralysis.

In the absence of sophisticated software for data management, personnel management and business administration, there is a risk of information loss and business collapse. Companies that do not set the direction of production will face the high costs of producing and developing two different products. An unresolved spatial layout and the absence of modern technology will limit the company; thus, the enterprise will start to lose the possibility of effective growth.

Companies engaged in atypical and highly atypical production must evaluate their direction and the future development of their order volumes. The digitization of the company and the introduction of innovative technologies in the non-manufacturing departments of these companies will always pay off.

The company should evaluate its production direction. The processing of the commodities of solid sawn timber and agglomerated materials requires additional technological specifics, machinery and spatial layout, while the division of development, production and machinery increases the costs of technology and company facilities. Respondents state: "The machinery originally used on a monthly basis should be replaced by the services of external suppliers, thus creating room for the modernization and innovation of their own facilities."

### 3.3. Implementation of Industry 4.0 in Medium-Sized Enterprises

In the case of introducing innovations in a medium-sized enterprise, the respondents agreed that the classic joinery workshop is losing its typical form (Table 6). Small woodworking machines are replaced by nesting milling centers, CNC machining centers with automatic part stacking, continuous lines connected by conveyors, continuous painting lines and other equipment. This realistic view of modern joinery production differs from the reality of today's vocational training for the various branches of joinery.

One of the major issues of concern of the respondents is the lack of preparedness of furniture manufacturing graduates from secondary schools and colleges. It often takes a year for them to understand the whole issue of digitalized production and modern technologies, while the employer expects the graduate to be immediately involved in production. This is unrealistic in relation to the classical joinery education, where modern joinery production is separated from classical production by the acquired competences. This practically makes it a separate discipline, which would be better identified in the teaching and when these requirements are included in the teaching.

Respondents also report that innovative technologies, both manufacturing and non-manufacturing, are essential to their business while increasing the efficiency of the overall operation. Specifically, respondents engaged in atypical production report efficiency gains ranging from 30%–40%. In enterprises that use batch production, labor efficiency gains of up to 50% over a 5-year period are observed. Efficiency gains are based primarily on reducing communication flows, errors and repetitive operations at all levels of the enterprise.

**Table 6.** Main results and recommendations (scored) for medium-sized enterprises and their subcategories (*n* = 10).

| Cat. | Subcat. | Example of a Real Company | Proposed Solutions | Benefits of Implementing Industry 4.0 | Risks of Non-Implementation within 5 Years |
|---|---|---|---|---|---|
| Medium-sized enterprise (up to 250 persons) | up to 150 persons, turnover up to 25 million EUR (very atypical production) | - Highly operational decision-making; - The management is usually handled by not very sophisticated software or by the classic personal assignment of work; - Long-term planning here is very challenging, highly operational management; - There are companies with advanced technical and technological backgrounds, but also companies with major shortcomings. | - Introduction of complex digitalization at the level of product parts; - Production planning, persons management using advanced complex software; - The life cycle of a project is planned by the software depending on its 3D data; - In production, the human member is always indispensable; - High demands on cloud storage and cyber protection. | - Global sophisticated software providing complex operations, data management, control and planning of all projects and persons in the company's flow; - The design department automatically generates all the documents for the production flow from the cyber twin; - Using QR codes, efficient data collection and record keeping; - Increasing labor productivity. | - The absence of a sophisticated software solution brings threats, data loss, cyber-attacks; - Increased demands on staff; - Constant pressure, stress and high responsibility; - Increased incidence of mental illness among workers; - Reduced fungibility, failure to meet deadlines, threat. |
| | up to 150 persons, turnover up to 25 million EUR (mass production) | - Multiple repetitive operations; - Long-term planning of production capacities, materials, human resources and more; - Business built on long-term contracts with customers; - Adapted facilities for the production of a given product; - Technical and technological equipment is more advanced than in atypical production companies. | - Digitalization of the entire communication flow management and technical and technological equipment; - Upgrading to the Intelligent Factory; - Shifting human resources from manufacturing to non-manufacturing positions; - Creating intelligent work cells of people, machines and software; - Optimization and automation and robotization of operations. | - Creation of new positions, filling by existing employees; - The management and operation of the company is provided by sophisticated software; - A society more resilient to individual failure; - More efficient production flow, flexible product changes, reduced costs, increased competitiveness; | - Inability to respond to price and quality competition; - Pressure of competition from countries with low production costs or using the attributes of the modern smart factory; - Impossibility of meeting delivery dates and prices; - Human factor failures often threaten the entire production flow of a company. |
| | up to 250 employees, turnover up to 50 million euros | - Modern high-capacity plants; - Experience with some automation and digitization; - High level of machinery, its constant modernization; - Companies are already addressing Industry 4.0 and its attributes; - Automation, robotization necessary for the operation of production. | - Using all the building blocks of Industry 4.0; - The smart factory and its cyber twin; - Big data collection and evaluation, use of IOT, IOS, HCPS; - Data monitoring and evaluation, for automated decision-making systems; - Created teams of people, software, machines. | - It ensures the running of the business; - Long-term planning and self-management; - Process automation; - Continuity and efficient operations of an intelligent enterprise. | - In the future, it will jeopardize the entire operation of the company and its infrastructure; - Without the use of the innovative technologies offered, it will not be possible to sufficiently and comprehensively manage traffic of this magnitude. |

### 3.3.1. Up to 150 Persons, Turnover up to EUR 25 Million (Very Atypical Production)

Businesses with atypical production face highly operational decision-making choices. Long-term planning is very challenging, and any material shortages or production errors will fundamentally disrupt the production flow and lead times, bringing the need for operational management. The division of labor, task control, project handover and communication across the company is usually handled by not very sophisticated software or by the classic personal allocation of work.

There are companies on the market with an advanced technical and technological background with modern machines, sawmills with chaotic storage systems and CNC nesting centers, but also companies that have major shortcomings in this regard, which reduce their production efficiency.

Proposed Solutions

Implementing the comprehensive digitalization at the product part level, production planning along with people management using an advanced complex software that provides fast real-time information transfer with the ability to view it from anywhere is required.

Stakeholders propose the implementation of sophisticated software to ensure the project life cycle. From the acceptance of the order, the project passes through the company, packs all the data (technical and economic) and is planned by the software itself with respect to its 3D data and the project life cycle, including the possibility of showing the customers themselves in the approval process and involving them in the production cycle.

The stakeholders emphasize: "In a company of this size, engaged in the modernization of enterprises, the human being as a member of production is indispensable. All machinery must be adapted to the needs of the final product and production flow." Here, man acts as a machine operator, an operator of equipment, and performs specific jobs that are difficult for machines to grasp.

High demands are placed on cloud storage and cyber protection, which must not be underestimated.

Benefits of Implementing Industry 4.0

Global sophisticated software providing complex operations, data management and control and planning of all projects and people in the company's flow, including production planning and real-time records of all orders and products, is required.

The design department creates a cyber twin of the product, which is a digital copy of the final product. In the process of data generation, cutting plans, data for CNC centers, electronic manufacturing processes, material ordering, creating a plan of expected production times and schedules of real production are automatically created. Subsequently, performance records of the machine and people, the material consumed and other data valuable to the company are collected in the manufacturing process.

Using QR codes or barcodes, the system is able to carry production documents directly from the design department and display them anywhere in the production or assembly.

There is a reduction in the number of technical and economic workers and an increase in labor productivity.

Risk of Non-Implementation within 5 Years

Without a sophisticated software solution, a company exposes itself to many dangers such as material shortages in production, data loss or cyberattacks.

Missing data records on corporate or external networks place increased demands on the existing communication flow of individual employees.

The management of a company, often by one person, and its possible reduced substitutability or failure to meet dispatch deadlines in the context of production outages and the need for high operational management are a threat in the long term. Constant pressure, stress and high responsibility are more likely to cause psychological illnesses among workers.

3.3.2. Up to 150 Persons, Turnover up to EUR 25 Million (Mass Production)

In companies engaged in batch and repetitive production, the production volume, material requirements, human resources and more can be planned more effectively over the long term. Companies often have long-term product supply contracts with their customers and may also have customized facilities for the production of specific types of products. The costs associated with producing a single part, minimizing waste, and efficient preparation strongly influence the efficiency of the business. In this subcategory of the firm, manufacturing becomes the heart of the enterprise.

The machinery tends to be more advanced than in atypical manufacturing companies. We can find, here, partial or full automation, modern banding centers, continuous paint booths and others.

Proposed Solutions

Multiple repetitive operations in serial production, digitized control of the entire communication flow, together with the modernization of plants and technical and technological equipment, create the ideal basis for a modern automated intelligent factory.

As part of the automation of operations, the need for manpower in production, which is carried out by machining, is decreased, with human resources being transferred to non-production areas. The formation of work teams takes place depending on the production cells. Teams are formed by human resources, machines or software addressing the optimization and automation of the necessary tasks.

Benefits of Implementing Industry 4.0

According to the respondents, innovative companies show an increased demand for skilled human labor in non-manufacturing parts. New jobs are being created so that employees do not have to worry about their jobs; they just have to adapt to the workload, which often becomes less physically demanding. Intelligent work teams in which humans and machines work together are more efficient and flexible, communicating with each other using display devices, and they are able to task each other.

The management and operation of the company is handled by information storage software, making the company more resilient to individual failure. There is the long-term planning of production, materials, logistics, etc. As a result, these innovations bring the possibility of better production flow, eliminating labor shortages and reducing costs.

Risk of Non-Implementation within 5 Years

The respondents point to the company's lack of development, reducing its ability to respond to price and quality competition and threatening the business in the long term. In the future, these enterprises will face more competition from markets with lower production costs, for example from eastern Europe or developing countries.

Delivery dates and product prices are decisive. Without innovation, the company will be severely compromised by more advanced and ready competitors using the attributes of the modern smart factory. There are often repetitive manufacturing operations of hundreds of units that can almost always be replaced by machines in these enterprises. Human failure often threatens the entire production flow of a company.

3.3.3. Up to 250 Employees, Turnover up to EUR 50 Million

Modern advanced production plants in serial or repetitive production that already have experience with some automation and digitalization are necessary. The machinery is at a high level, constantly being renewed and modernized. Companies are already addressing the question of the individual attributes of Industry 4.0. They are using cloud storage, cyber protection, partially or comprehensively collecting digital information, etc.

Companies engaged in mass production also have elements of automation and robotics that are effectively applied and essential to the operation of the company.

Proposed Solutions

Using all the building blocks of Industry 4.0, specifically, we can talk about the implementation of the smart factory and the digital twin of the enterprise for the long-term production and human resource planning.

The company continuously monitors and collects data that serve as the basis for automated decision-making systems.

Benefits of Implementing Industry 4.0

The attribute of full Industry 4.0 integration ensures the operation of the business, long-term planning and self-management. This ensures the creation of intelligent production blocks, creates teams of people, software and machines and maximizes process efficiency.

Example: a customer creates a product in the e-shop in pre-prepared libraries (cyber twin of the product), the design can be consulted by an in-house architect, ordered and then paid for. In the next step, automated processes in the IoS interface, the automatic ordering of materials, the creation of production documentation, production planning, the start of a partially or fully automated production process (all repetitive operations performed by several people are eliminated) all take place. The item passes through a production flow where 3D data is displayed, and the efficient control and planning of the entire production process takes place. In the production flow, big data are collected by the IoT, stored on cloud repositories where they are evaluated and distributed to the machines and equipment in the IoS interface. Thanks to HCPS, machines and people work together to create efficient structures.

Risk of Non-Implementation within 5 Years

Underestimating the importance of the global perspectives offered by Industry 4.0 can jeopardize the entire operation of a company and its infrastructure in the future. The opportunities and innovations that the market offers must be exploited, monitored and continuously integrated into the company's strategy. Otherwise, data, contracts and quality employees may be lost.

Without the use of the innovative technologies offered, it will not be possible to sufficiently and comprehensively manage traffic of this magnitude.

The digitalization and complexity of the project flow from the confirmed order to assembly with the overall display of documentation and data collection of products or individual product parts represents a major shift and streamlining of process management. Corporate stakeholders recommend incorporating software solutions that allow the evaluation and simulation of the flow of orders through the company and incorporate the above aspects. The company that has implemented this measure reports that, in the three years since implementation, production has increased by up to 50%, with an investment of hundreds of thousands of euro. In production, it is about managing people and projects, displaying production data, reading barcodes, collecting data and all the information that is used to evaluate the information that makes it possible to manage individual processes.

### 3.4. Implementation of Industry 4.0 in Large Entesrprises

Intelligent large-scale furniture manufacturing plants (Table 7) are a harbinger of the future that all industries are heading toward. Respondents agreed that the introduction of innovation increases labor productivity and reduces the need for direct production staff, which is increasingly difficult to find in the market. These enterprises are also subject to the general requirements set for medium-sized enterprises, e.g., the need for a training response. Here, it is possible to fully integrate all the attributes of Industry 4.0 and implement sophisticated solutions as in the previous groups. Enterprises usually have sufficient know-how, employing a specialist to deal with various subsidy titles. The company also has a specialist focused on development and innovation.

**Table 7.** Main results and recommendations (scored) for large enterprises and their subcategories (*n* = 5).

| Cat. | Subcat. | Example of a Real Company | Proposed Solutions | Benefits of Implementing Industry 4.0 | Risks of Non-Implementation within 5 Years |
|---|---|---|---|---|---|
| Large company (more than 250 employees, turnover more than €50 million) | built by gradual modernization | - Limited by the spatial layout of the businesses; <br> - The mechanical and technological progress here is much worse than that of a "greenfield" company. | -Application of all the building blocks of Industry 4.0; <br> - Application of CE principles; <br> - Waste utilization, energy sources; <br> - Supporting the next generation of the workforce. | - Full automation and digitalization of the company; <br> - Connecting the company, increasing efficiency and competitiveness; <br> - The amount of innovation and efficient layout is influenced by the spatial layout of the enterprise; <br> - Efficient use of waste. | - Underestimation of certain attributes during the design, threatens the efficiency of production, lack of employees, etc.; <br> - Lack of stability in competition; <br> - The threat is lack of staff, high energy costs, large amounts of funds tied up in warehouses, loss of data and more; <br> - Failure to address the energy situation and partial self-sufficiency threatens the business with high costs. |
| | built on a "green field" | - Enough finance, know-how; <br> - Implementation on a "green field" with minimal layout restrictions; <br> - The attributes of Industry 4.0 can be effectively implemented; <br> - Efficient layout of operations and machines; <br> - Fully automated large capacity plants; <br> - Serial production. | | - Construction on the "green field" is not limited by the layout of the production hall; <br> - The most efficient production lines ensuring automatic movement of parts, their machining, assembly and packaging; <br> - Finding new ways to increase productivity. | |

### 3.4.1. Built by Gradual Modernization

Large enterprises, created by the gradual growth of their capabilities and capacities, often take several decades to take shape. Companies of this kind are often limited by the space available in their production facilities. Halls are often renovated, as old buildings are limited by support columns, low ceilings or narrow halls. Mechanical and technological progress is far inferior to that of a greenfield company. Companies of this type have sufficient experience and know-how to innovate and develop the business. The efficiency of the whole company is increased by approximately 50% compared to the original operation.

### Proposed Solutions—For Both Subcategories of Large Enterprises

According to the stakeholder, the application of all the building blocks of Industry 4.0 is necessary. The use of circular economy (CE) principles in waste management with the possibility of using a cogeneration unit and waste treatment is required. With the gradually increasing demands for cleaner production, these steps will be necessary in the future. Supporting a new generation of workers by working with secondary and higher education institutions and retraining workers is necessary.

### Benefits of Implementing Industry 4.0

The modern automated factory and digital interconnection of the company will bring very significant efficiency gains. It all depends on the range of products, the seriality and the layout of the company's premises. The layouts allow for an efficient arrangement of woodworking machines along with the fully automatic movement within the overall production flow. Minimizing waste and using it efficiently will help reduce energy costs.

### Risk of Non-Implementation within 5 Years—For Both Subcategories of Large Enterprises

A company of this size must take all the attributes of Industry 4.0 into account when designing or reconstructing its operations. If it underestimates some of the attributes, it risks failure, inefficient production, a lack of employees, etc. In a competitive struggle, an

insufficiently innovative company will be at risk. In the near future, the enterprise may be replaced by a company with a clear vision and goal. Interviewees of large enterprises point to the threats of a lack of skilled workers competent in Industry 4.0, high energy costs, large amounts of funds tied up in warehouses, loss of data and others. The lack of preparation in partial energy self-sufficiency, for example, by waste treatment or solar panels, threatens the enterprise in times of high energy fluctuations.

### 3.4.2. Built on a "Green Field"

Businesses that have sufficient finance and know-how will build a factory on a "green-field" site that is not constrained by the layout of already existing buildings and structures. In this variant, all the attributes of Industry 4.0 can be effectively planned, integrated, fulfilled and controlled. The given layout of the operations and machines can be adapted directly to the targeted production. Here, we can talk about fully automated large-scale units engaged in mass production.

#### Benefits of Implementing Industry 4.0

If a company plans to build a greenfield company, it is not limited by the layout of the production halls. This is where the smart factory concept with all the attributes of Industry 4.0 comes into play. The goal of the investors is to design the most efficient production line, ensuring both the automatic movement of manufactured parts and their machining, surface treatment, assembly and packaging. Companies are constantly looking for new ways to increase productivity. The waste generated can be a major burden, but its recovery in energy production reduces its threat and the costs associated with running a business. Decision-making processes within the CE will bring substantial efficiencies to the entire company.

According to stakeholders of large companies, the amount of investment depends on the level of added technological value and the complexity of the processes. The return on the entire investment should be approximately five years and the investment can range from hundreds of thousands to millions of euro. However, the added value of the entire production flow usually increases by more than 50% within five years in large enterprises compared to non-innovative enterprises.

### 3.5. Practical Implementation of Industry 4.0 in the Furniture Industry

From the experience of the respondents who are involved in the overall implementation of Industry 4.0, a model of the complete implementation of Industry 4.0 was established on the basis of structured interviews.

Rising costs and competitive pressures require a high degree of automation, intelligence and flexibility. Industry 4.0 is a solution capable of coordinating the flow of information between all departments in a company through technologies and networks that facilitate communication between the process participants (machines, people and devices), making their daily tasks easier and eliminating repetitive processes.

A smart factory (Figure 2) is a data-driven enterprise where intelligent devices can perform computation, communication or precision control. The day-to-day operation of a smart factory relies heavily on the maturity of the software (3D printing, cloud computing platform, manufacturing execution system, virtual reality, smart logistics, etc.) and technology equipment (robotic, self-cutting centers, glue centers, CNC machine tools, etc.).

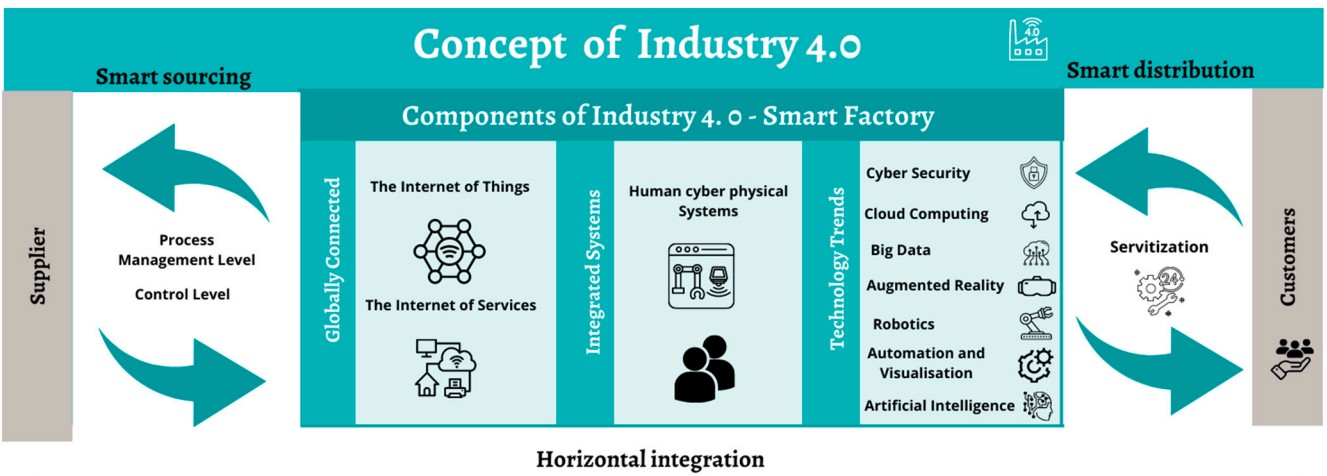

**Figure 2.** The new concept of Industry 4.0 connecting the smart factory closely with customers and suppliers.

In terms of project management, the entire furniture production process depends on the size of the company and the specifics of the production. The furniture manufacturing company itself should be divided into departments (sections) through which individual projects move toward an efficient goal. The section must have clearly defined rules, structure and job descriptions for each employee, so that each employee knows the scope, goal and extent of their work.

The entire smart grid is divided into two parts. The transfer of information in the company is undertaken by machine-to-machine (M2M) applications that change the dialog between man and machine. The second part is the exchange of information in the factory, which takes place through the Internet of Things. The IoT provides big data collection and system communication with transfer to cloud storage. During the actual collection, on the network or in cloud storage, the data are evaluated through the Internet of Services, which forms a global internet connection between systems and services. Furthermore, decentralized information is distributed on the basis of which the systems can make decisions. Thanks to these technologies, HCPS, driven by computer algorithms from physical sources, provides self-organizing structures and enables interaction between the different participants in the process within artificial intelligence systems. With increasing digitalization, the risk of cyberattacks is also increased, making cybersecurity an integral part of furniture operations.

To organize a smart factory, it is necessary to have an overview of all the materials involved in the production of furniture. It is necessary to know, at all times, what needs and requirements production has and how to satisfy these requirements in the most efficient way. Automatic inventory management together with software classifies all materials (raw materials, semi-finished and finished products) and provides all this information via lists to the responsible personnel, as well as manages the logistics and outputs of the final product. It is also linked to the calculation of purchase requirements and automatically generates orders to suppliers, respecting the production and storage criteria used in the company. Through servitization and trade management, a complete service is provided to customers and suppliers, including the quality control of the incoming material and waste processing. The vision for the future is the indirect involvement of the customer in the production process, e.g., through an intelligent e-shop, display of current production statuses, etc.

Different groups of attributes can be intertwined within the maturity of an enterprise and the following tables indicate the basic distribution of their capabilities. The technologies used should always be one level more advanced than the company's current need. All innovative processes moving toward Industry 4.0 technologies streamline and optimize the production time, reduce any potential errors and allow physically demanding and

sometimes dangerous tasks to be transferred from man to machine. A human team member solves a complex problem and an AI team member handles less complex activities or more repetitive operations. This creates a competitive advantage, ensuring the optimal use of the resources, as well as environmental responsibility.

## 4. Discussion

The results reveal several insights and best practices regarding the effective implementation of Industry 4.0. In this study, we focused on furniture companies that are involved in Industry 4.0 and used guided interviews with project managers to find out the current status and benefits of Industry 4.0.

In the following, the most important similarities are discussed and compared with the current state of research using the person, organization and technology model.

The analysis of the interviewed enterprises and the guided interviews revealed that there is a relatively low level of readiness of SMEs for the specific use of Industry 4.0 compared to large enterprises engaged in mass production. This finding can be explained by the fact that large enterprises have a much higher availability of resources for the use of technology, have the know-how and understand the importance of innovative technologies. In their size and manufacturing focus, they are practically indispensable and have the space to focus on strategically oriented activities. This finding is in line with previous studies dealing with the implementation of Industry 4.0 [6]. According to the findings of the respondents, in the case of a large enterprise (more than 250 employees, turnover more than EUR 50 million), Industry 4.0 is an essential component. If an entity has sufficient funding, know-how and builds a greenfield factory, all the attributes of Industry 4.0 can be met more easily than for a company that is gradually modernizing and is limited by the different spatial layouts of its facilities. Large and medium-sized mass-production companies can apply all the attributes of Industry 4.0 to their portfolio, thereby increasing their production and capacity while reducing their costs. Interviewees point out that "Software solutions, machinery and companies supplying innovative turnkey Industry 4.0 solutions are mainly specialized for large enterprises with mass or repetitive production". This is confirmed by the article analyzing furniture companies. The authors point to the lack of supply chain and hardware and software services for SMEs [36].

One of the many practical outcomes and practical impacts will be to connect SMEs in the implementation of Industry 4.0 solutions, which will bring competitive advantages to these businesses such as lower costs, higher yields and the rational use of green and sustainable renewable resources without the need for their own high investments. The practical experience of the respondents shows that as a business grows from a micro-enterprise disposition to a large enterprise, it undergoes major development, innovation and internal changes.

According to the findings, it is essential that the transformation of operations leads to simple yet flexible industrial robotic cells. A large part of the wood manufacturing industry needs to revamp its production systems and develop new manufacturing technologies or software solutions [18]. An important factor for the success of the application of the individual parts of a smart factory is to start with the limited implementation of individual cells rather than a comprehensive reconstruction. It is important to proceed only when the work process is understood by the workforce, i.e., human resources [27]. Some entities considering the application of robotic and automated lines consider modern technology as a tool to solve most of their problems. Open communication within the company and education are necessary to avoid misunderstandings and possible failures [17,19,27]. This is confirmed by the respondents, who point to the need for strategic planning of the entire implementation with clear objectives for each section of the company. Repeated communication and clarification of the intentions must also take place with employees, for whom it is important to identify with the company's objectives and the implementation of Industry 4.0. They must come to understand that automation and robotics help them to increase efficiency and replace particularly demanding and monotonous work. Through

the process of servitization and customer involvement in the production process to address environmental issues, it can either directly generate economic benefits or indirectly, through environmental or operational performance, increase customer satisfaction. This route can also maximize the volume of all downstream production and supply processes [44,45]. Unfortunately, the very meaning of servitization is not properly understood by all Industry 4.0 companies. Even if they perform certain services in this regard (e.g., design department, e-shop, etc.), there is no targeted integration and value creation [32].

Stakeholders interviewed unanimously point to two of the most significant barriers to the implementation of Industry 4.0 in the wood-based materials processing sector (furniture manufacturing). The first factor is small-scale production, which requires very flexible mechanical cells in the manufacturing plant itself. This is accompanied by high programming requirements reducing profitability and efficiency. Second, the machining of solid wood requires a very sophisticated approach, as machining parameters must be constantly adapted to the characteristics and processes of the wood being processed. The processing of raw timber requires the use of single-purpose machines (e.g., dimensioning saws, trimming saws, horizontal milling machines and presses), which use up space and staff capacity. The large amount of waste, energy and human labor consumption thus becomes a burden not only for the company itself but also for the environment [46,47].

The next most frequently cited barrier to respondents taking any action was the problem of investing time and money in development. This also applies to other sectors such as forestry or textiles [48–50]. Before deciding to invest in Industry 4.0, business owners should see examples that these practices are profitable and provide many other benefits. Current business representatives consider innovative solutions to be difficult to manage financially and unprofitable. Implementing an innovative solution is undoubtedly more difficult than continuing with the traditional way of working. However, the implementation of Industry 4.0 can be adapted to the capabilities and needs of a given company.

According to the respondents, gaps in the business supply chains on both the buyer and supplier side are central to the implementation of Industry 4.0, with the biggest problem being the introduction and co-operation with various entities that do not use modern technologies and sharing and communication with these partner entities is difficult. Nowadays, the concepts and tools of business management are rapidly expanding. Top management is, therefore, focused on implementing the entire Industry 4.0, including the business and supply chain [35]. A well-integrated and -managed supply chain is considered a powerful strategic and logistical "weapon" that is difficult to imitate and provides a long-term competitive advantage [38,39].

All of the managers interviewed in the case of large and medium-sized enterprises see a general problem in the fact that service providers offering a combination of hardware and software are mostly specialized for large companies. There is practically no company that offers a small enterprise services in a comprehensive solution of technical and technological equipment and sales of certain "know how" addressing the complete services of a production and non-production nature. Here, we come up against a fundamental fact, which is the lack of information about possible solutions in the whole furniture manufacturing sector. Another problem is the incompatibility between the machines providing the different manufacturing operations and the software controlling or managing them, and the intercommunication between different software machines from different manufacturers. There is a great opportunity for companies to offer business analysis and implement a solution that mediates communication between the warehouse, design software, human resource management software and accounting software and across other operations of the company [7]. The aim is to streamline the communication flow, display the necessary data and eliminate repetitive operations applicable in wood-processing companies, which is also confirmed by Jasinska [51]. The interviewed enterprises engaged in development often employ a specialist who programs the product. Many companies still do not see the potential that lies, for example, in cooperation with specialist outsourcing entities [27]. It is important to note that, for some processes, it is more efficient to use the services of

specialized IT entities than to carry out the process with in-house IT staff [52], such as data management and cyber protection, software interconnection, etc. When implementing, it is advisable to proceed in parts; it is recommended to complete one part first and then work on the next part.

One of the main principles of Industry 4.0 is the creation of operational production plans and a reduction in the physical inventories. Given the current situation and the status of the supply chain, this is now very problematic and a revival would be advisable. Across the industry, supplies of materials needed for production are delayed or stopped altogether as a result of the COVID-19 pandemic [53,54], the Russian–Ukrainian war, high inflation and across-the-board price increases for all products and services. The absence of a wide range of materials, fittings and electronics supplied from Asian countries is a current issue, where transport times play a major role. This problem highlights the unpreparedness of society and industry in the broader spectrum of all the supply and production flows, forcing companies to build up stocks of materials and components in which large amounts of money are tied up. With the gradual increase in transport prices along with the green deal, producers of scarce goods produced outside the EU will be forced to move production to European countries as well. By reducing transport times and ensuring the self-sufficiency of European countries, more rapid deliveries would also be achieved. At the same time, however, there are specific differences in the conditions of individual countries in the furniture industry [55].

Furniture companies should use the opportunity associated with green energy promotion to improve furniture waste recycling mechanisms to minimize energy consumption [21]. One of the main factors that causes resource overload is the global system based on linear flow of materials and energy, which causes the depletion of natural resources and the generation of large amounts of waste [56]. The appropriate solution is a circular model based on a circular economy (CE), which allows for sustainable development. Another option is to use waste as a source of energy in the company, for example, for cogeneration units producing electricity that will then be used in production, which will significantly reduce the production costs of the company and can have a significant effect, especially in the current energy crisis. This is confirmed by some companies that are already producing electricity from their residue.

Stakeholders cite staff shortages, both in production and technical management positions, as the most common risk of non-implementation. In these cases, the enterprise is often managed by one person and its possible reduced substitutability or failure to meet shipment deadlines in the case of a production failure and the need for high operational management are threats in the long term. This risk is most evident in medium-sized and large enterprises, where Industry 4.0 technologies make production more efficient, and the number of technical and economic staff is reduced. All four industrial revolutions have had a major impact on work in terms of education. According to the interviewees, there is currently a large deficit of competent and skilled workers (workforce) in the European labor market with experience in the field and competence in innovative technologies, as confirmed by a study from Spain [57,58]. Along with the requirement to retrain employees, the workforce needs to meet sophisticated production conditions and acquire soft skills [59,60]. The automation and industrialization of plants will also have an impact on human resources in the sense of Human Resources 4.0. In the long term, jobs lost will be replaced by jobs that meet the needs of the future market [10,61]. An aspect that raises concerns about the implementation of Industry 4.0 is the current education system setup. With the advent of digitization and robotics, production processes as such are changing fundamentally and the education system should respond immediately [62]. Outdated joinery plants have nothing in common with modern production. School graduates are usually fresh out of apprenticeship and do not master digitized processes, which significantly hampers the development of companies. Respondents strongly perceive this problem and cite cooperation between schools and technically advanced companies as a solution. Here, pupils would undergo an apprenticeship that would help raise awareness of a wide range of opportunities, for

example, as in the automotive industry. Some of the enterprises interviewed are already implementing this concept and are positive that it is of great importance for the students' education. This subsequently applies to the enterprises for jobs as graduates of the schools involved.

### 5. Conclusions

The new innovative technologies of Industry 4.0 are key to the future development of the furniture sector. Methodical support for managers who are the key users of these technologies is essential in applying the new Industry 4.0 trends to the furniture sector. In this study, we were focused on furniture companies that are involved in Industry 4.0 and used guided interviews with project managers to find out the current status and benefits of Industry 4.0. The study proposed a methodological framework in each of the key areas for implementing Industry 4.0 in companies, broken down by the company size. Furthermore, recommendations for future research in this sector emerged from this study. In particular, the focus was on new trends in furniture manufacturing and the effective implementation of Industry 4.0 within different company size categories. According to the findings of the respondents, it is necessary to involve small companies in the process of implementing Industry 4.0 in this sector in addition to large companies where Industry 4.0 is commonplace, as they represent a significant potential for implementing Industry 4.0, especially in terms of supply chains. The different approaches to the implementation and technologies of Industry 4.0 need to be simplified, especially for SMEs. They are mainly limited by a lack of financial resources, knowledge and business organization. Therefore, it is necessary to use only partial attributes that small enterprises can realistically use. In conclusion, all the interviewed managers agreed that innovative technologies of production and non-production types are essential for their companies, where their applications have undergone rapid shifts, increasing the efficiency of the entire operation in the range of 30%–50%, reducing the communication flow, error rate and repetitive operations at all levels of the enterprise and ensuring the efficient use of renewable resources in line with the Sustainable Development Goals (SDGs).

Further follow-up research can help to clarify other aspects of Industry 4.0 implementation. Using different samples of enterprises with respect to, for example, the nationality of the wood and forest industry sector, can elucidate any nuances that exist. In addition to this study, a survey among different stakeholder groups, especially among supply chain representatives of enterprises, will also be conducted. It is also appropriate to focus the Industry 4.0 communication strategy on supply chains and SMEs, which represent a significant potential for the success and development of Industry 4.0 in the furniture sector.

To summarize the results of this study, it can serve as a basis for addressing strategic decision-making in project management in the application of Industry 4.0 in enterprises; it can also be successfully applied in other sectors as the principles for each category of enterprise will be very similar. To succeed in the competition in the long term, enterprises must continue to evolve.

**Author Contributions:** Conceptualization, L.Č., R.S., T.Č.; methodology, R.S.; validation, L.Č., T.Č.; formal analysis, L.Č., R.S.; investigation, T.Č.; sources, L.Č.; writing—drafting, L.Č., R.S., T.Č.; writing—revision and editing, R.S., T.Č.; visualization, T.Č.; supervision, R.S.; project administration, R.S.; fundraising, R.S. All authors have read and agreed to the published version of the manuscript.

**Funding:** This research was funded by the non-project research funds of the Faculty of Forestry and Wood Technology of the Czech University of Life Sciences, Prague.

**Acknowledgments:** The authors wish to thank all the stakeholders who took part in this research and made it possible. The authors also thank Harvey Cook for proofreading and editing the article.

**Conflicts of Interest:** The authors declare no conflict of interest.

## Abbreviations

| | |
|---|---|
| AI | Artificial intelligence |
| AR | Augmented reality |
| HCPS | Human–cyber–physical system |
| IoT | Internet of Things |
| IoS | Internet of Services |
| CE | Circular economy |
| CNC | Computer numerical control |
| EU | European Union |
| SMEs | Small and medium enterprises |
| SDGs | Sustainable Development Goals |
| NACE | Nomenclature statistique des activités économiques dans la Communauté européenne |

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
