# Peer review of "The Potential of Smart Factories and Innovative Industry 4.0 Technologies—A Case Study of Different-Sized Companies in the Furniture Industry in Central Europe"

_forests, doi:10.3390/f13122171_

Round 1

Reviewer 1 Report

1. Introduction and Literature review section are fall to provide exact novelty, finding, and problem definition. Thus, authors are advised to recreate those section very carefully In the introduction, please refer to and review the literature on the issues of the industry 4.0. I recommend the following articles:
Swain, M., Zimon, D., Singh, R., Hashmi, M. F., Rashid, M., & Hakak, S. (2021). LoRa-LBO: an experimental analysis of LoRa link budget optimization in custom build IoT test bed for agriculture 4.0. Agronomy, 11(5), 820.
Baghizadeh, K., Zimon, D., & Jum’a, L. (2021). Modeling and optimization sustainable forest supply chain considering discount in transportation system and supplier selection under uncertainty. Forests, 12(8), 964.
Rana, A., Rawat, A. S., Afifi, A., Singh, R., Rashid, M., Gehlot, A., ... & Alshamrani, S. S. (2022). A Long-Range Internet of Things-Based Advanced Vehicle Pollution Monitoring System with Node Authentication and Blockchain. Applied Sciences, 12(15), 7547.
etc. 2. Need proper explanation of each figures and tables, such that reader can understand your work. . 3. Limitation and future extension should be explained properly in the conclusion section with proper references. 4. The method is interesting but without sufficient literature it reads like a report from a research firm. So, what makes this study unique and how are the findings advancing the field? Or what are they contributing to practice? 5. Please develop a discussion section. Good Luck!

Author Response

Dear reviewer,

the article has been revised and supplemented with your suggested changes, such as:

- The introduction has been revised and added

- The literature review has been expanded, including the articles you suggested.

- Explanations of figures and tables have been made.

- The discussion has been substantially revised according to your request.

Thank you for your insightful comments.

Reviewer 2 Report

The subject of the paper is interesting and in line with the aims and scope of the Journal. However, the paper is poorly structured and written. It does not provide any significant scientific contribution. It lacks a proper literature review, and almost all sections have serious flaws. More detailed comments are provided below.

1.     The abstract is not written well. It is very hard to understand due to the long, never-ending sentences. In addition, it lacks descriptions of the methodology, main results, conclusions, and main scientific contributions.

2.     The introduction is not written well. It does not provide the main results, conclusions, and main scientific contributions of the paper.

3.     The paper lacks a proper literature review. Some literature is reviewed in the introduction, but this was not done properly. A literature review does not cover all the main topics of the paper (e.g. industry 4.0 technologies and their applications, in general, but also in the furniture industry – there are plenty of papers covering this topic, etc.). In addition, one of the main results of this review should be the identification of the research gaps that this paper will cover. This has not been done.

4.     There are many inconsistencies in the paper. For example, the authors state that „...the following main factors were identified that affect the...“ (lines 176-177) but then provide non of them. Which main factors do they refer to?

5.     The authors did not provide any specific information on the number and structure of the investigated companies, nor the information about the number, background, experience, and expertise of the interviewed managers.

6.     One of the main drawbacks of the paper is the lack of any original or innovative methodology. A simple application of structured interviews can hardly be seen as a significant scientific contribution. Actually, the paper hardly has any significant scientific contribution. There are some interesting conclusions drawn from the interviews, but as we don’t know anything about the sample (size, structure, etc.) we can hardly adopt these conclusions as universal.

7.     Part of the paper under the title “Discussion” is not the discussion. This paper lacks discussion. Everything under the heading “Discussion” looks more like the background and the literature review. The authors did not discuss how the results they obtained in the study can be interpreted from the perspective of previous studies. Discussion should clearly and concisely explain the significance of the obtained results to demonstrate the actual contribution of the article to this field of research, when compared with the existing and studied literature. The authors also failed to highlight the limitations of the paper as well as the theoretical and practical implications.

8.     The Conclusion is not written well. The authors provided conclusions of the interviewees, but not their own conclusions based on the results of their study. In addition, the authors did not provide any future research directions. There should be at least 3-5 solid future research directions that would be interesting to the majority of the Journal readership.

9.     Thorough English revision is required. There are plenty of spelling, grammar, syntax, and style errors. There are also a lot of long and hard-to-understand sentences that need to be broken down into several sentences (e.g. the sentence in the abstract between lines 15-21).

10.  There are certain technical issues:

a)     The paper is not formatted according to the Instructions for Authors (provided template). For example, the font size for the headings is too big.

b)    The sub-heading 1.1. is too extensive. Headings should be concise and informative.

c)     There should be at least a couple of sentences between headings of different levels (e.g. between section 3 and sub-section 3.1).

d)    Tables are not formatted according to the Instructions for Authors.

e)     The tables are too extensive. There is no need for this amount of text in the tables. Tables should be informative and supported by the main text. For example, table 4 contains more text than the actual text in this sub-section 3.1.3.

f)     Acronyms/Abbreviations/Initialisms should be defined the first time they appear in the main text (for example „EU“ is defined after its first appearance in the main text). Check the rest of the abbreviations. There are more mistakes (for example „CM“ is used for Cloud Storage).

g)    References in the reference list are not formatted according to the Instructions for Authors.

h)    Volume and page numbers missing for the reference [4], the volume number missing for the reference [6] and [9], etc. Check the rest of the references.

Author Response

Dear reviewer,

thank you for your comments. Due to the number of comments, the entire article has been revised and supplemented with your suggested edits, such as:

-The abstract has been edited, but due to the small possible word count, it cannot address the entire stated issue,

-Changed the entire introduction.

- Modified the literature review (in more detail).

- Revised introduction, discussion and conclusion.

- Language proofreading by a native speaker as per your recommendation.

- Due to problems with the original template, it was inserted into the new template.

A revision of the text was made by a native speaker, which should eliminate grammatical errors.

Thank you for your insightful comments. 

Reviewer 3 Report

The paper discusses more or less some guidelines to various categories of companies regarding the modernisation in the scope of industry 4.0. The paper is comprehensible. However, there are numerous issues which need to be solved comprehensively by changing the paper tremendously.

The biggest issue is vagueness regarding the novelty of this paper. What is new in this paper in the scope of the previous work? What are the main findings of the paper? What will the reader learn from the paper?

This paper contains results but methodology is actually missing. Data and exact methodology for the results is not sufficiently described. Which companies were included in this case study? How was the interview performed? There is statistical analysis missing. How were the guidelines derived? The paper has to be changed significantly.

Regarding results: Authors state some general vague statements but justification/analysis/numbers/source is missing. Some examples: “The machine, which was originally used on a monthly basis, should be replaced by the services of external suppliers, which will create space for modernisation and innovation of its own facilities.” and ”Without a sophisticated software solution, the company is exposed to many dangers, such as a lack of material in production, loss of data or cyber-attacks.”, “where there is an increase in production by up to 50% with investments in the hundreds of thousands of EUR”. How were such claims derived?  Why 50 % and not 10 %? Same applies to other claims over the paper. Source of data and/or methodology is missing for such statements.This paper has to have more methodological structure.

Regarding grammar: The paper is readable but some sentences are very long and hard to follow, starting with lines 15 to 21. Some sentences have very low information value and are redundant: “A new phenomenon of Industry 4.0 is the introduction of Smart (intelligent) factories, which are a key element of Industry 4.0.”

Through the paper, there are numerous typos. Some examples.
line 35: “,,”
line 147: ”(obstacles. Advantages”
line 151: “which were …”
line 152: “sociologists This“
line 337: “of EUR However”

Author Response

Dear reviewer,

the article has been completely revised according to your request and therefore the change is not only in the form of revisions but completely the whole article.  We have tried to incorporate your suggestive comments.  Thus, the introduction, literature review, methodology, discussion and conclusion have been revised and the changes requested by you have been added. A native speaker also revised the text, which should eliminate grammatical errors.

Thank you very much for your insightful comments.

Round 2

Reviewer 1 Report

I have no more comments.

Reviewer 2 Report

The authors have made an astonishing effort to correct their paper according to the comments from the previous review round. The quality of the paper has been significantly improved. I suggest the acceptance of the paper in its present form.

Reviewer 3 Report

The paper has been sufficiently improved. Methodology is more clearly described. I have no more comments left, only some small typos:
 - "up to 25 persons e"

 - Supplementary materials: Figure S1: title;